# Dynamic Wetting Properties of Silica-Poly (Acrylic Acid) Superhydrophilic Coatings

**DOI:** 10.3390/polym15051242

**Published:** 2023-02-28

**Authors:** Sevil Turkoglu, Jinde Zhang, Hanna Dodiuk, Samuel Kenig, Jo Ann Ratto, Joey Mead

**Affiliations:** 1Plastics Engineering Department, University of Massachusetts Lowell, Lowell, MA 01854, USA; 2Department of Polymer Materials Engineering, Shenkar College, Ramat Gan 5252626, Israel; 3U.S. Army Combat Capabilities Development Command Soldier Center, Natick, MA 21005, USA

**Keywords:** surface wetting, superhydrophilicity, nanocomposite coating, dynamic spreading

## Abstract

Superhydrophilic coatings based on a hydrophilic silica nanoparticle suspension and Poly (acrylic acid) (PAA) were prepared by dip coating. Scanning Electron Microscopy (SEM) and Atomic Force Microscopy (AFM) were used to examine the morphology of the coating. The effect of surface morphology on the dynamic wetting behavior of the superhydrophilic coatings was studied by changing the silica suspension concentration from 0.5% wt. to 3.2% wt. while keeping the silica concentration in the dry coating constant. The droplet base diameter and dynamic contact angle with respect to time were measured using a high-speed camera. A power law was found to describe the relationship between the droplet diameter and time. A significantly low experimental power law index was obtained for all the coatings. Both roughness and volume loss during spreading were suggested to be responsible for the low index values. The water adsorption of the coatings was found to be the reason for the volume loss during spreading. The coatings exhibited good adherence to the substrates and retention of hydrophilic properties under mild abrasion.

## 1. Introduction

The wetting behavior of a solid surface can be determined by the water contact angle. There are two important extreme wetting conditions that have been studied by researchers for many decades: superhydrophobicity [1,2,3] and superhydrophilicity [4,5,6]. Surfaces that are completely wetted by water, are extremely water-loving, or have a static water contact angle of less than 10° are considered superhydrophilic surfaces and have been well-studied [7,8]. These surfaces are preferred for a number of practical applications, such as antifogging [9,10], antifouling [8,11], and self-cleaning [12,13].

While superhydrophilic surfaces are defined by a static contact angle of less than 10°, the contact of a liquid drop with any surface results in a change in the contact line of the liquid drop until the liquid–solid–vapor system reaches an equilibrium state of less than 10 degrees. Dynamic wetting, the spreading behavior of a liquid on a solid surface, occurs in numerous everyday situations and is important for many practical applications, such as paints, antifogging, ink-jet printing, textile dyeing, metal or glass anticorrosive coatings, lubrication, gluing, plant treatment, and cosmetology [14,15,16,17]. The dynamics of a liquid drop prior to reaching the equilibrium state are heavily affected by surface chemistry, surface topography, and liquid properties [18].

Many theories have been proposed for wetting dynamics, based either on hydrodynamics or on a molecular theory of the contact line [19]. In hydrodynamic theories, the relevant parameter is the capillary number (Ca), which is generally regarded as the dimensionless contact line speed. Ca can be defined as the ratio of viscous forces to surface tension forces, as given in Equation (1), where U is velocity, µ is viscosity, and γ is the surface free energy of the solid–liquid [20]. Most of the analytical descriptions in the literature were reported by assuming that Ca ≤ 0.1, as it yields the simple power law relationship [21].
(1)$$\mathrm{Ca}=\frac{µ\;U}{\gamma }$$

An increase in the capillary number results in a dynamic contact angle increase: $\theta_{d}={\left(\mathrm{Ca}\;\right)}^{1/3}$ [22,23].

The early- and late-stage spreading behavior of a liquid drop on a highly wettable, smooth, solid surface has been studied for decades, and various power laws, such as a spreading radius that changes with time, have been proposed to predict the spreading characteristics [24]. Tanner (1979) first defined the spreading of a small liquid drop on a smooth and completely solid wetting surface. Tanner’s law (the hydrodynamic theory) constructs a power law of the spreading radius, $R_{d}\sim t^{1/10}$ and $\theta_{d}\sim t^{-3/10}$, whereas the molecular theories lead to $R_{d}\sim t^{1/7}$ and $\theta_{d}\sim t^{-3/7},$ where *R* is the dynamic drop radius, $\theta_{d}$ is the dynamic contact angle, and *t* is the time [25,26,27,28]. These theories apply to flat surfaces and high-viscosity liquids.
(2)$$r\left(t\right)\sim R{\left(\frac{\gamma t}{\mu R}\right)}^{1/10}$$

Surface roughness plays a significant role in dynamic behavior. The dynamic spreading of different liquids has been studied for highly ordered structures prepared using various techniques, including lithography and deep reactive ion etching [18,24]. A high-viscosity silicon oil is widely used to study dynamic spreading, as it completely wets the surface. Some studies have shown experimental data that are in good agreement with Tanner’s law [29,30].

However, Sawicki [31] studied the spreading rates of PDMS liquid droplets with different viscosities and showed that the $R_{d}\sim t^{1/10}$ relationship was not followed for lower-viscosity liquids. McHale et al. [32] studied the dynamics of spreading of silicon oil on lithographically produced surfaces with aspect ratios larger than four and found that the spreading dynamics of their surfaces followed power law behavior but showed a change from Tanner’s law in the dynamic contact angle to $\theta_{d}\sim t^{-3/4}$ from $\theta_{d}\sim t^{-3/10}$ due to the surface roughness. Courbin et al. [18] studied the dynamics of silicon oil on a chemically homogeneous surface made from a square pattern of PDMS micropillars prepared using soft-lithography methods. They demonstrated that the spreading radius had a power law dependence on time, $R_{d}\sim t^{1/2}$, and a change in the initial drop shape affected the exponent of the power law. Ruijter et al. [19] studied the spreading of a droplet of di-*n*-butylphthalate (DBP) on a poly(ethyleneterephthalate) (PET) substrate. They showed that a $R_{d}\sim t^{1/7}$ and $R_{d}\sim t^{1/10}$ relationship was followed for (0.008–4.5 s) and (4.5–15,000 s) time frames, respectively. The resulting different power laws were attributed to the different wetting models when the system was near or far from its equilibrium state. 

While there have been studies on ordered, patterned surfaces, there are very few studies focused on low-viscosity liquids or the spreading dynamics of randomly rough surfaces. Xu et al. studied the spreading of silicon oil on stainless steel surfaces with different degrees of roughness ($R_{a}\sim \;0–25$ μm) and formed via spark erosion [33]. Their study showed that the contact line mobility of a drop was lower on a rougher surface. Herminghaus [34] developed an analytical expression to predict the spreading and adsorption of liquid on randomly rough substrates.

For low-viscosity liquids, such as water, the spreading process on a rough surface exhibits a $R_{d}\sim t^{1/2}$ spreading dependence, as predicted by the Washburn Equation, developed for cylindrical capillaries during the first 30 ms period [35,36]. Kim et al. [24] studied the dynamics of both water and silicon oil liquid drops on regular micropatterns. They found that the radius of the drop followed $R_{d}\sim t^{1/4}$, which is different from Washburn’s law, and the spreading rate increased with the initial drop volume and surface tension and decreased with increasing viscosity.

The dynamic wetting behavior of randomly rough surfaces is important for superhydrophilic coatings, which are attractive for practical applications because of their facile manufacturing. Thus far, studies focused on the dynamics of wetting for randomly rough surfaces have presented either modeling without any experimental verification or experimental results without modeling. These studies have only shown a relationship between the droplet base radius vs. time or contact angle vs. time. However, the mechanism controlling the dynamics of wetting on randomly structured superhydrophilic coatings has not been fully explored.

In this paper, the dynamic wetting properties of randomly structured polyacrylic acid/nanosilica superhydrophilic rough (micro- and nano-level) coatings, were studied both experimentally and through modeling to explain the effects of the coating composition and roughness on the dynamic wetting behavior of the superhydrophilic coatings.

## 2. Experimental Section

### 2.1. Materials and Chemicals

Poly (acrylic acid) (PAA) (Mw = 450,000) and colloidal nanosilica particles LUDOX TM-40 (40% wt. SiO_2_ suspension in water, with an average particle size of 22 nm, pH of 9.0, and surface area of 110–150 m^2^g^−1^) were purchased from Sigma-Aldrich [10,37]. Plain glass microscope slides (75 × 25 mm) were used as a substrate (Fisher Scientific Company, Cat. No. 12-550-A3). Deionized water (electrical conductivity of 0.05 µS/cm) was used for preparing the water-based solutions and for all rinsing activities [38,39].

### 2.2. Preparation of Coatings

The superhydrophilic coatings were prepared following our previously described method [40]. First, PAA aqueous solution was prepared by dissolving the PAA powder in deionized water (1% wt.) and stirring (350 rpm) at 85 °C for 12 h. Then, PAA aqueous solution was slowly added into a predetermined amount of hydroxylated SiO_2_ colloidal suspension (LUDOX TM-40) while stirring at 350 rpm for 45 min to prepare PAA/SiO_2_ dispersions. The composite PAA/SiO_2_ coatings were prepared by dipping the bare glass samples into the different PAA/SiO_2_ suspensions with their different solid (PAA/SiO_2_) contents. Uncoated glass slides were cleaned with isopropyl alcohol and deionized water and then dried with nitrogen prior to the coating process. The coated glass slides were dried at room temperature for 5 min before heating. The coated samples were heated to 120 °C in an oven for 3 h and then cooled to room temperature for 12 h. A schematic representation of the superhydrophilic coating fabrication process is demonstrated in Figure 1.

The water droplet (with a volume of about 2 μL) spreading dynamics on the synthesized superhydrophilic coatings with different roughnesses were investigated by changing the silica concentration in suspension from 0.5% wt. to 3.2% wt. while keeping the particle to binder (PB) ratio, the ratio of the silica to PAA, the same at 19:1. The binder refers to the polymer, which in this case is PAA. In this study, we aimed to investigate the effect of viscosity on the dynamic wetting properties. To do so, we took the SS0.5 as a starting point and then increased the concentration. All prepared formulations are given in Table 1.

### 2.3. Characterization

The water contact angle measurements were carried out using a high-speed imaging technique with the sessile drop method (Drop Shape Analyzer—DSA100-KRÜSS GmbH, Hamburg, Germany). Water droplets of 2 µL in volume were dropped onto the coating surfaces to be tested under the conditions of an ambient temperature and atmosphere. The equilibrium water contact angles (EWCA) were defined as the contact angles when the water droplet ceased to advance. Both the EWCA and dynamic wetting properties of the synthesized coatings were assessed by monitoring the water contact angle and droplet base diameter (illustrated in Appendix A) from the initial milliseconds to several intervals (at least three times) for each sample. The value of the static water contact angle was determined with an accuracy of ±1°.

Images of the coated samples were taken using a field-emission scanning electron microscope (JSM 7401F, JEOL Inc., Peabody, MA, USA) at an electron energy of 2 to 5 kV. The samples were coated with a nanometer sputtered thin gold film to avoid surface charging.

Surface topography measurements were performed using an atomic force microscope (AFM) (PSIA 100) with a scan rate of 0.5 Hz. Roughness measurements were carried out by scanning a 20 × 20 μm^^2^^ area with a non-contact AFM tip. Image processing and analysis were performed using XEP software and Image J software.

Attenuated total reflection (ATR) spectra of the coated samples were obtained using a Fourier-transform infrared (FTIR) spectrometer (Nicolet FTIR6700, Thermo Fisher Scientific Inc., Waltham, MA, USA). The spectrometer had a single-reflection diamond ATR accessory with a DTGS detector and KBr beam splitter operating at room temperature. Spectra were obtained over 4000–500 ${\mathrm{cm}}^{-1}$, with an average of 64 scans and a spectral resolution of 4 ${\mathrm{cm}}^{-1}$.

Transmission measurements were conducted with an ultraviolet–visible spectrophotometer (UV-Vis, Hitachi U-2910 spectrophotometer). Percent transmission was recorded in the range from 400 nm to 700 nm.

The mechanical durability of the coatings was examined with the Taber abrasion test following the G195-13a ASTM standard. Cylindrically shaped CS-10 abrasion wheels that produced a medium abrading action using a 250 g load were used for the abrasion test. Changes in the equilibrium water contact angles were measured after the abrasion cycles.

A tape test was performed using ASTM D3359-17, where a lattice pattern is formed with a cutting tool on the coated substrate. Pressure-sensitive tape was applied to the lattice pattern and removed carefully. The adhesion was evaluated qualitatively on a scale from 0 to 5.

## 3. Results and Discussion

The water droplet (with a volume of about 2 μL) spreading dynamics on the superhydrophilic coatings were investigated by changing the solids’ (binder and silica) concentration in suspension from 0.5% wt. to 3.2% wt. while keeping the silica to binder ratio the same (19:1) in the dry coating (silica concentration constant (91% wt.)). As the concentration of silica increased in the suspension, the viscosity increased, affecting the final coating characteristics (thickness, roughness, etc.), which, in turn, affected the spreading dynamics.

The coatings in this study were all superhydrophilic, showing an equilibrium contact angle of <10 degrees. The prepared coatings showed good long-term stability in their wettability (superhydrophilic), with no change in the equilibrium water contact angle after being kept at room temperature for at least 2 months. The coatings showed an initial dynamic change in contact angle, reaching equilibrium within 5 s. The dynamic contact angle was monitored for all coatings, and the images are illustrated in Figure 2. It was noted that the contact angle gradually decreased in the first few seconds. The equilibrium WCA measurements (Figure 2, images at 5s) showed that for all coatings (95% silica in the dry coating), superhydrophilic behavior was observed, even at a very low (0.5 %) wt. of silica in the suspension.

The droplet spreading behavior on the fabricated coatings was monitored with a high-speed camera. The droplet base diameter (D) was measured at different time intervals (t) by analyzing the captured high-speed camera images. The relationship between D and t is demonstrated in Figure 3 in a log–log plot. As shown in Figure 3, a straight line is obtained using a logarithmic presentation. The spreading droplet diameter (D) has a power law relationship with the spreading time (t), which conforms to the published literature [18]. However, the slope of the double logarithmic relationship is around 0.05, which is significantly lower than those reported for other coatings [24,41]. The power law indices for the various coatings are listed in Table 2. The indices were similar for each coating, with values of approximately 0.05. A similar power law index (0.0569) was obtained for a PAA coating from the graph presented in Appendix A.

Low power law indices were also reported in the literature [35]. The reduced wetting rates were attributed to porosity and surface roughness, which slow the wetting process by partially pinning the three-point contact line (TPCL) [42].

To further characterize the reasons for low power law indices, the synthesized coatings were analyzed for their roughness and the drop volume change during spreading.

For the roughness, the coating topography was analyzed by AFM, as demonstrated by the 2D and 3D AFM images in Figure 4. As claimed by Garnier et al. [43], surface asperities greater than 0.16 μm are thought to significantly reduce the wetting rate. The synthesized coatings in this study possess surface asperities and agglomerates, which are randomly structured. The AFM images were analyzed by Image J software to obtain the asperities’ size in the coating. The sizes of the resulting asperities in the coatings are listed in Table 3. As is evident in the table, as the wt.% of nanosilica in the suspension was increased from 0.5 to 3.2, the viscosity of the coating increased, and the asperities’ dimensions increased from a nanometer to a micrometer level.

Parameters of the roughness statistics of the coatings were obtained through AFM analysis and are listed in Table 4. The data show that increasing the % wt. of silica in suspension increased the roughness factor value (the ratio of the actual surface area to the projected surface area), which was related to an increase in asperity size. Table 4 also demonstrates that both the $R_{\left(RMS\right)}$ and $R_{a}$ values increased with the increase in the wt. % of silica in the suspension.

Although the change in surface roughness should have changed the spreading behavior [18], no significant change was found in the experimentally derived power law index values. While most literature shows that roughness affects drop spreading, the obtained roughness statistics indicate that surface roughness did not play a significant role in the rate of droplet spreading in this system.

Traditional theory enables the determination of the rate of change in the base diameter of a spreading droplet when the volume of liquid is constant [15,32,33,44]. The effect of volume change on the spreading behavior plays a significant role in the spreading behavior of a droplet. Mc Hale et al. [32] reported that when the volume loss exceeds 1% of the initial drop volume, the power law indices are reduced. Therefore, the decrease in droplet volume in this study was anticipated to contribute to the low power law indices and reduction in the droplet spreading speed.

Thus, the volume of the droplet during spreading was monitored using a high-speed camera. The droplet volume was calculated using Equations (3) and (4), where $\Phi \left(\theta_{d}\right)$ is a geometrical term that relates the base radius of the drop, $R_{d}$, to its volume [27].
(3)$${\left(R_{d}\right)}^{3}=\left(\frac{3\;Vol}{\pi }\right)\Phi \left(\theta_{d}\right)$$
(4)$$\Phi \left(\theta_{d}\right)=\frac{{sin}^{3}\theta_{d}}{2-3cos\theta_{d}+{cos}^{3}\theta_{d}}$$

The relationship between the calculated droplet volume and the spreading time is demonstrated in Figure 5. The data in Figure 5 indicate that the droplet volume decreases with the spreading time. A volume loss of more than 25% during spreading is exhibited for each coating. Thus, it is hypothesized that the volume loss during drop spreading on these superhydrophilic coatings significantly affects the spreading behavior and the observed low power law index.

It is postulated that droplet volume decrease during spreading may be the result of adsorption, as the coating formulation consists of hydrophilic polyacrylic acid and hydrophilic nanosilica, which can interact with the water drop and adsorb water. To further understand the volume decrease, the volume change of a pure polyacrylic acid coating (on a glass substrate) was evaluated. As illustrated in Appendix A, a volume decrease of more than 15% was observed for the neat PAA coating. This volume decrease is attributed to the adsorption of water by PAA due to chemical and thermodynamic interactions.

Since the water volume decrease on the PAA coating (around 15%) was smaller than the volume decrease on the coatings containing nanosilica (>25%), it can be concluded that surface roughness should be considered an additional plausible reason for the volume decrease. To further investigate the volume decrease during spreading, the morphology of the coatings was analyzed.

The microstructure of the coatings in Figure 6 shows that the coated surfaces are dense, crack-free, and uniform. The coated surfaces containing silica nanoparticles exhibited a good nanoparticle packing and distribution. As Figure 6 illustrates, the increase in the wt.% of silica in the suspension increased the agglomerate size in the final coating. The SEM images for the various coatings display rougher coatings with the wt.% increase in the nanosilica in the suspension.

It should be emphasized that the nanosilica concentration in the starting suspension and its resultant viscosity play significant roles in the resulting roughness (agglomerate size) of the final coating. As the silica nanoparticle content in the initial suspension increases, it significantly increases the suspension viscosity [45,46], with the obvious consequence of larger agglomerates in the final dry coating.

High-magnification images of the SS 0.5 coating are presented in Figure 7. Individual silica nanoparticles of around 22 nm can clearly be seen. The good dispersion of the nanoparticles is evident, with improved optical and mechanical properties. Figure 7 shows that both the individual silica nanoparticles and the agglomerates that are composed of the silica nanoparticles are well-packed and distributed in the dry coating.

Cross-section SEM images of the coatings are illustrated in Figure 8, showing that the coatings had different thicknesses, which ranged from the nano- to micro-level. As expected, the coating thickness increased with increased viscosity.

The FTIR analysis of the samples is shown in Appendix A. The resultant spectra indicate that a broad band appears on the spectrum of the coating at around 3365 ${\mathrm{cm}}^{-1}$, which is related to OH stretching and hydrogen bonding. The spectrum also shows SiO_2_-related peaks at 1025 ${\mathrm{cm}}^{-1}$ (Si–O–Si stretching), 760 ${\mathrm{cm}}^{-1}$ (Si–O bending), and 903 ${\mathrm{cm}}^{-1}$ (Si–OH). The peak at 1635 ${\mathrm{cm}}^{-1}$ is likely due to the C=O stretching of carboxylic acids and O–H deformation of carboxylic acids and alcohols. There are also peaks appearing at 1122 ${\mathrm{cm}}^{-1}$, which could be attributed to C–O stretching vibrations, and at 824 ${\mathrm{cm}}^{-1}$, related to C–H bending. The tabulated peak assignments of functional groups are given in Appendix A. When the spectra of the uncoated bare glass slide and PAA are taken into account [40], one can conclude that these data show no indication of new peaks in the coating. However, there were significant differences between the coatings with different suspension concentrations in terms of the Si–OH band. These variations are most prominent in SS0.5. This could be attributed to the glass support due to the lesser thickness of the SS0.5 coating.

### Performance Analysis of the Coatings

The abrasion resistance characteristics of the prepared nanocomposite coatings are presented in Figure 9. All coatings showed an increase in EWCA after the Taber abrasion test. It is also evident that increasing the abrasion cycle number resulted in further increases in EWCA. The coatings reached the lowest EWCA value of 38° after 30 abrasion cycles under a constant load of 250 g. The increase in EWCA has been attributed to changes in the coating surface caused by the Taber abrasion test, which affected the surface roughness and morphology [47]. The coatings maintained appreciable hydrophilic characteristics throughout up to 30 cycles of abrasion. The images of the coatings after 30 cycles are given in Appendix A and illustrate that the majority of the coatings remained intact after Taber cycling. However, some parts of some of the coatings were lost, exposing the glass, after several abrasion cycles. Although the water contact angle measurements were taken on the abraded area where the coating appeared to be intact, the exposed glass support could have also affected the contact angle values. Measurements were not taken on the exposed glass areas. The work of adhesion ($W_{A}$) was calculated after every 10 cycles of the Taber test, according to Equation (5), where cosθ is the equilibrium water contact angle measured after every 10 cycles and $\gamma_{L}$ is the surface tension of the liquid (water), which is known in the literature ($\gamma_{L}$= 72.8 mN∕m) [48,49]. The results are presented in Appendix A, revealing that the work of adhesion value decreases by approximately 10% after 30 Taber cycles. This results in a change from superhydrophilic to hydrophilic behavior, which is illustrated in Figure 9.
(5)$$W_{A}=\gamma_{L\;}(1+cos\theta )$$

The adhesion of the coating to the substrate can be measured with the tape test [50,51]. According to the ASTM D3359 standard, the adhesion is rated from 0B to 5B, where 5B represents the highest rating for the tape test durability [52]. In this study, all formulated coatings showed a 3B–4B adhesion level, as demonstrated in Table 5. Since the adhesion level determines how well the coating adheres to the substrate, we concluded that the superhydrophilic coatings show good adhesion between the substrate and the coating.

The transparency of the coatings was characterized by UV–Vis spectra in the spectral range of 400–700 nm. The optical transmittance of the prepared coatings is illustrated in Figure 10. The transparency was drastically decreased with the increase in the solids’ loading (binder and silica) in suspension, imparting a higher viscosity and, thus, a higher thickness and roughness. The decrease in transparency was associated with an increase in the thickness and surface roughness of the coatings.

## 4. Conclusions

Superhydrophilic thin coatings, composed of silica nanoparticles and PAA with various random roughnesses, were prepared by dip coating into different concentrations of solids (silica nanoparticle and PAA) in water suspensions. Because of the different viscosities of the suspensions, the coatings had different thicknesses and morphologies. The coatings were studied with respect to their wetting characteristics. The spreading dynamics of water droplets were investigated by relating the droplet base diameter to time. The different coatings showed similar (but very low) power law indices. Droplet volume decrease was considered to be the main reason for the low power law index values. The adhesion properties for all of the coatings were similar and demonstrated good adhesion to the substrate. The abrasion resistance of the superhydrophilic coatings (on glass substrates) exhibited good retention of the hydrophilic attributes for up to 30 abrasion cycles. The coatings showed a moderate increase in EWCA after Taber abrasion cycles. The UV–Vis spectra demonstrated that the coating transparency was dependent on the coating thickness and roughness levels, with the maximum values exceeding 90%.

## Figures and Tables

**Figure 1 polymers-15-01242-f001:**
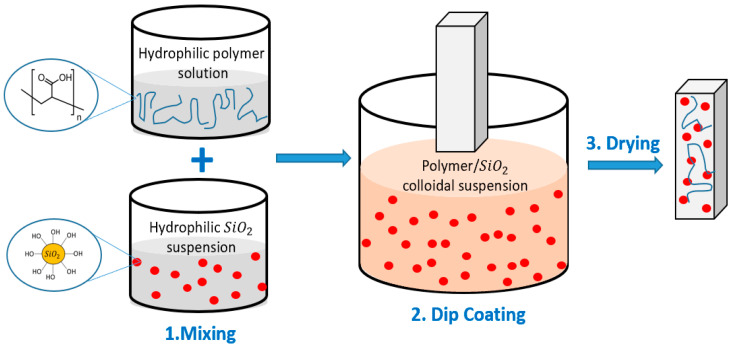
Coating process schematic for superhydrophilic coatings.

**Figure 2 polymers-15-01242-f002:**
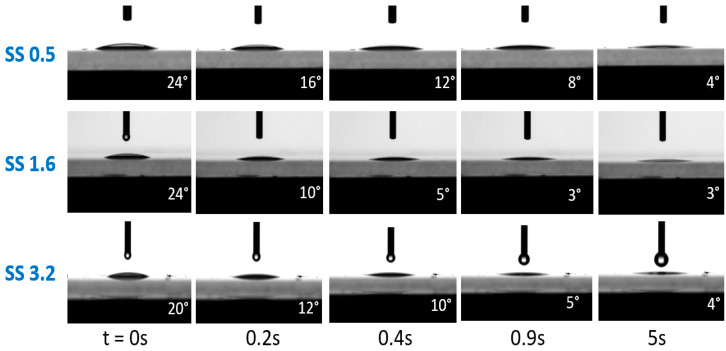
Water contact angle with respect to time. The diameter of the needle is 0.5 mm.

**Figure 3 polymers-15-01242-f003:**
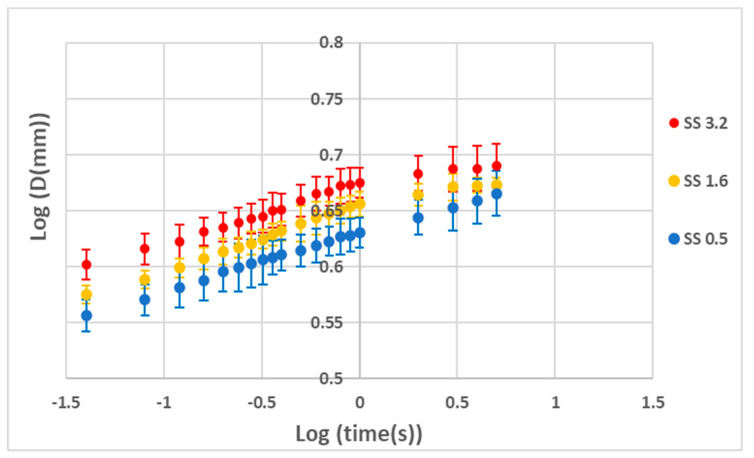
Plot of Log(D) vs. Log(t) for coatings with different % wt. of silica in suspension.

**Figure 4 polymers-15-01242-f004:**
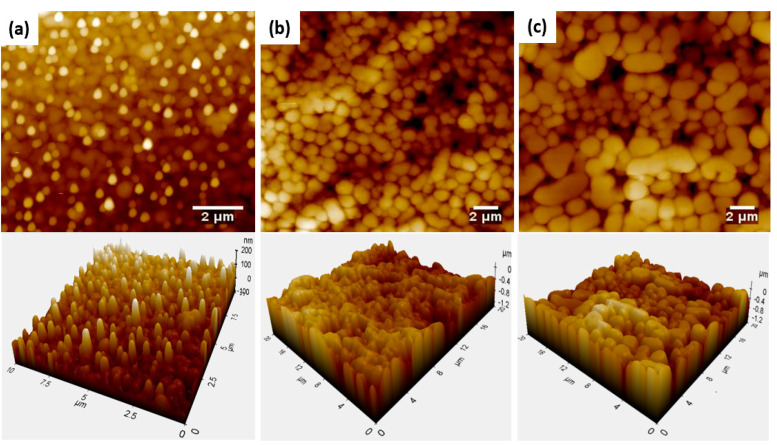
AFM topography of the superhydrophilic coatings with % wt. of silica in suspension of (**a**) 0.5, (**b**) 1.6, and (**c**) 3.2.

**Figure 5 polymers-15-01242-f005:**
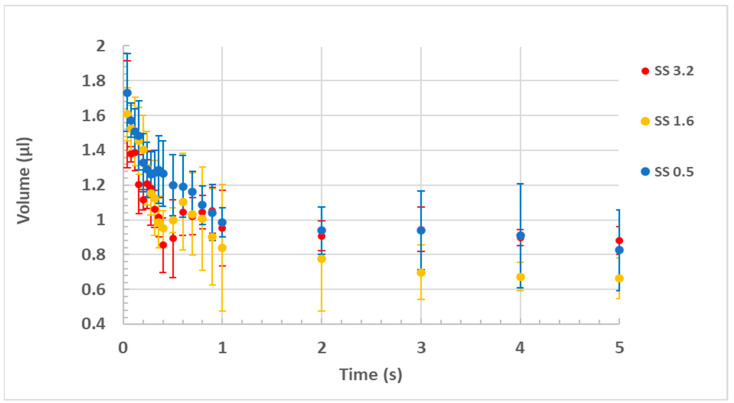
Volume change with time for coatings with different % wt. of silica in suspension.

**Figure 6 polymers-15-01242-f006:**
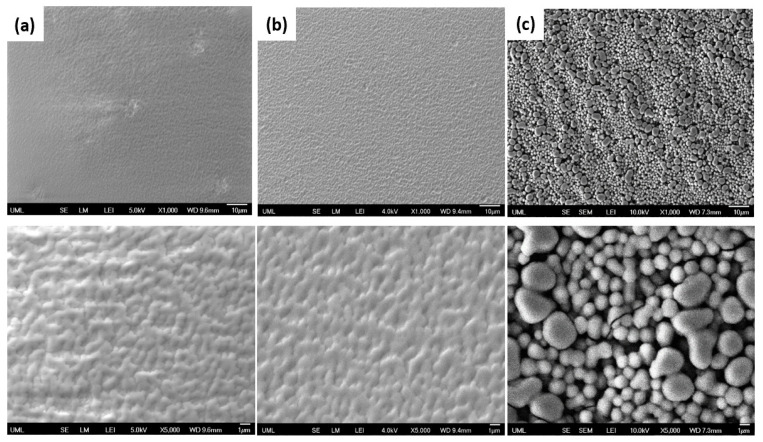
SEM low-magnification (1000× on top) and higher-magnification (5000× on bottom) images of the superhydrophilic coatings with % wt. of silica in suspension of (**a**) 0.5, (**b**) 1.6, and (**c**) 3.2.

**Figure 7 polymers-15-01242-f007:**
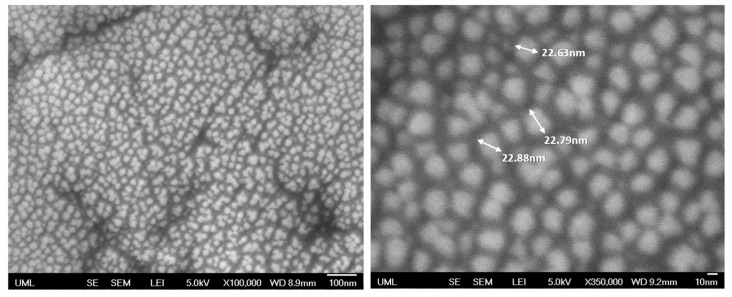
SEM high-magnification (100k on **left** and 350k on **right**) top-view images of SS 0.5.

**Figure 8 polymers-15-01242-f008:**
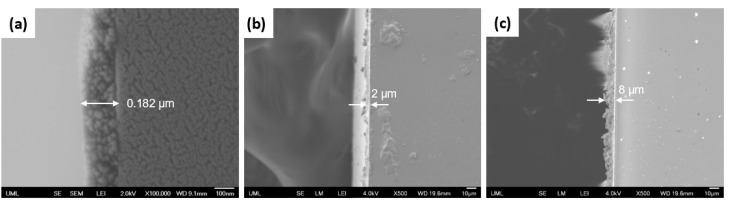
SEM cross-section images of coatings with % wt. of silica in suspension of (**a**) 0.5, (**b**) 1.6, and (**c**) 3.2.

**Figure 9 polymers-15-01242-f009:**
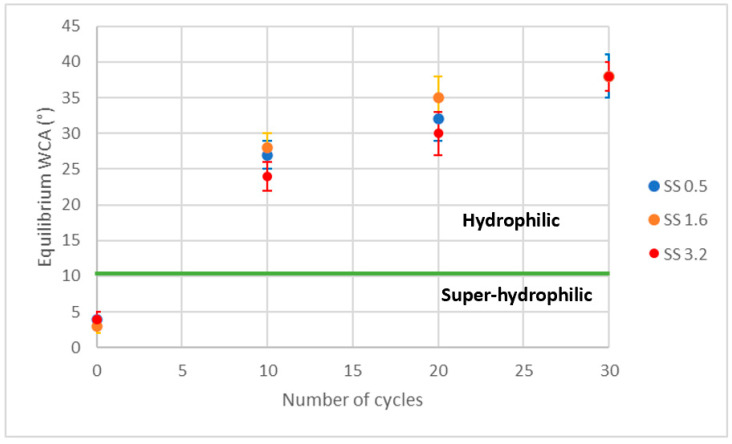
Taber abrasion test data.

**Figure 10 polymers-15-01242-f010:**
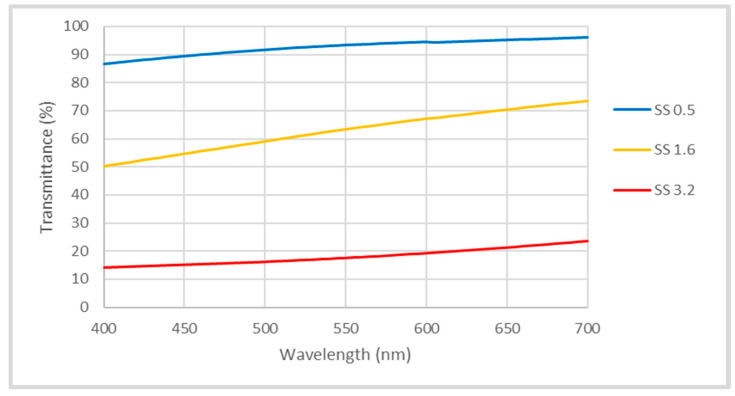
The transmittance (%) vs. wavelength graphs of different coatings.

**Table 1 polymers-15-01242-t001:** Superhydrophilic coating formulations.

Sample Set	% wt. of Silica in Dry Coating	% vol. of Silica in Dry Coating	Particle to Binder (PB) Ratio	% wt. of Silica in Suspension
SS 0.5	95	91	19:1	0.5
SS 1.6	95	91	19:1	1.6
SS 3.2	95	91	19:1	3.2

**Table 2 polymers-15-01242-t002:** Power law indices for coatings with different % wt. of silica in suspension.

Sample Set	% wt. of Silica in Suspension	Power Law Index
SS 0.5	0.5	0.051
SS 1.6	1.6	0.050
SS 3.2	3.2	0.045

**Table 3 polymers-15-01242-t003:** Coating asperity sizes (Image J Software).

% wt. of Silica in Suspension	0.5	1.6	3.2
**Asperity size (μm)**	0.60 ± 0.18	1.16 ± 0.17	1.69 ± 0.86

**Table 4 polymers-15-01242-t004:** Roughness statistics for the studied compositions.

% wt. of Silica in Suspension	Roughness Factor (r)	$R_{rms}$ (nm)	$R_{a}$ (nm)
0.5	1.03	48 ± 5	58 ± 3
1.6	1.21	223 ± 2	169 ± 2
3.2	1.24	244 ± 5	189± 4

**Table 5 polymers-15-01242-t005:** Adhesion test results.

Sample Set	Adhesion
SS 0.5	4B-3B
SS 1.6	4B-3B
SS 3.2	4B-3B

## Data Availability

The data presented in this study are available on request from the corresponding author.

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
