# Peer review of "Dynamic Wetting Properties of Silica-Poly (Acrylic Acid) Superhydrophilic Coatings"

_polymers, 2023, doi:10.3390/polym15051242_

Round 1

Reviewer 1 Report

Manuscript ID: polymers-2207815

Title: Dynamic Wetting Properties of Silica-Poly (acrylic acid)

Superhydrophilic Coatings 

Dear Authors,

 Thank you for the opportunity to read your article. I found the topic is interesting and fundamental. Generally speaking, there are some results presented in order to capture some trends but they can be better described and discussed. The conclusion and results need more clear explanation and/or more precision. I suggest that this article will be revised before re-submission. I recommend its major revision at this state. My comments are below and in the main text marked yellow. I hope they will be helpful.

Good luck

 “Keywords”

No keywords, why???

Abstract

Please correct very carefully English style.

Why such concentration is used?

Explain “power low coefficient”??

 “1. Introduction”

Unfortunately, this abbreviation Ca is confusing with the calcium symbol, is it not worth changing this abbreviation for e.g. Cn???

Besides all symbols should be written in mathematic style.

According to nomenclature n should be written Italics: di-n-butylphthalate (DBP)

In general, literature of other authors is very limited. A lot of own works are cited, it is not very professional. Please consider citing more literature.

Please consider clearly defining your work goals in terms of their applicability. 

“2. Materials and Methods”

Give conductivity of double distilled water, it is very important for potential readers?

22 nm it is small value, from what kind of method was measured??, what about aggregation process, please explain.

Please explain why 2mL was used for contact angle measurements, it seems too small value because according to wettability theory and thermodynamics give big error. Give some examples of similar studies of other authors.

Droplet advancing diameter??? Spreading droplet diameter???

Advancing and receding contact angle should be…

In my opinion, it is a big confusion of several concepts from the theory of the wetting process, please read about the basis of this methodology and hysteresis of the wetting angle.

Wettability and Stability of Naproxen, Ibuprofen and/or Cyclosporine A/Silica Delivery Systems, Colloids and Interfaces, 2022, 6(1), 11

Theory of hysteresis of Prof. Chibowski

The obtained roughness statistics indicate that surface roughness does not play a significant role in droplet spreading behavior in this system.

 I cannot agree with this, most of the literature data contradicts it, probably the methodology of measuring the contact angle failed, too small a drop volume, too few repetitions, etc.

Physicochemical characteristics of chitosan-TiO2 biomaterial. 2. Wettability and biocompatibility, Colloids and Surfaces A: Physicochemical and Engineering Aspects, 2021, 630, 127546

Page 11

Work of adhesion should be calculated. See some papers.

“Changes in wetting properties of silica surface treated with DPPC in the presence of     phospholipase A2 enzyme”       Applied Surface Science 256 (24) (2010) 7672–7677

“3. Results”->3. Results and discussion

In general, the results should be describe and discussed with more details and fair point of view. Please consider revising this section significantly. Please see my detail comments below.

“4. Conclusions”.

 Precise conclusions, it is too general. Remember some readers read only conclusion, must encourage the reading of the entire work. This phenomenon is unclear but …. we have an idea how to verify it

or probably the mechanism is ... You can't give up like that right now

Literature part should be corrected.

Literature must be carefully selected, the current one is very random, old or very old papers, inadequate to the subject of the article, the paper mentions superhydrophilic surfaces and the literature is quite the opposite about superhydrophobic surfaces, why??? It is not professional

Many errors, style of journals should be the same

Please polish English more.

Final conclusion:

Major revision

Reviewer 2 Report

The manuscript presents some data on the wetting behavior of silica-based coatings having random surface morphology. Such investigation is useful, since “there are very few studies focused on the spreading dynamics of randomly rough surfaces as well as low viscosity liquids.” From the Introduction and this work, it appears that the power law coefficient cannot be predicted and should be measured every time, depending on several parameters.

Below I gathered a few observations.

 -In Figure 1, the representation of hydrophilic silica is exaggerated. Please correct the formula.

 -Please define “binder”.

 -Figure 3: I may be wrong, but my impression is that two lines could describe better the plots (before and after 1s).

 -Garnier et al (35) –please correct the typing.

 -In eq. (4) pay attention to subscript “d”.

 -Since neat PAA was taken as reference for the droplet volume decrease, the Power Law Coefficient should be determined /mentioned in this case, too.

 -“Moreover, there were no discernible differences between the coatings from different suspension concentrations and the spectra were dominated by the polyacrylic acid.” In this case, this phrase is not accurate. The bands in PAA are not quite visible (which is normal given the low content in PAA), while there is significant difference between samples regarding the Si-OH band (much more pronounced in SS05, but this could be due to the small thickness of this sample, and attributed to the glass support).

 -The abrasion tests indicate strong modification of the surface (roughness, morphology?) after 10 cycles. Probably, the silica particles at the surface are removed, and the contact angles reflect the exposure of the glass support. By the way, the contact angle of the used glass slides should be mentioned. Also, images of the samples after the tests should be shown, to prove that there is still a coating after 30 cycles.

 -The tape tests: “it is concluded that the superhydrophilic coatings remain intact following the tape test”. The results are the same as reported in [33] for the same composition, and are somehow in agreement with the abrasion tests. Since 5B is the highest score, the 3B-4B cannot indicate “intact” coating, but medium –high adhesion, i.e. some “delamination” may occur in this test, too. Please verify.

 -A comparison of mechanical resistance and transparency between these samples and those reported previously [33] should be introduced. The new coatings were intended to follow the wetting properties, and most of the Results and Discussion section is focused on explaining the low values of the Power Law Coefficients obtained. However, a conclusion is difficult to reach for only three samples. Maybe the authors could find a way to treat the other samples, reported in [33], in the same manner, at least for confirmation. 

Reviewer 3 Report

The innovation and novelty of this study needs to be highlighted in the introduction, how to differentiate these Silica based coatings with the previous studies. Any additional benefits and features?

In 2.2 Preparation of Coatings, what’s the weight percentage of the PAA powder in water solution?

The SEM images are sharp and high quality, however, in Figure 6C, it’s hard to see the continuous polymer film which should come from the PAA phase, only silica aggregation phase is sharp and clear.  Explanation is needed.

What’s the mechanism that leads to the good performance in abrasion resistance test? What’s the origin of good adhesion to the substrate?

“Superhydrophilic thin coatings composed of 95% silica nanoparticles” Confusing sentence in the conclusion, where is the 95% comes from?

Round 2

Reviewer 1 Report

The authors write that they calculated the adhesion based on the contact angle, the equation is entered into the manuscript,
but the data is not included.
There is only adhesion data, the old ones based on tests.
Please add the correct one.

Author Response

We thank the reviewer for the comment and believe our presentation of the data was confusing.

The data was placed in the supporting information (Table S2), but we added a few sentences in the manuscript to describe it. 

“From Table S2 it is seen that the work of adhesion value decreases approximately 10% after 30 Taber cycles.  This results in a change from supehydrophilic to hydrophilic behavior, which is seen in Figure 9.”

We also changed the wording in the following paragraph to avoid confusion.

“Adhesion of the coating to the substrate can be measured by the tape test. 50, 51

Reviewer 2 Report

The authors responded most of my questions. However, there are still issues to discuss.

1) Since PAA is soluble in water, studying the wetting behavior is somehow strange. Such a coating would simply dissolve on long exposure to water (please comment on this in the applicative context). It is true that the amount of PAA is very low, but nevertheless, it should provide coherence to the silica layer. The very low power law coefficient seems to be due to PAA, in addition to increasing amounts of silica, which is hydrophilic too (an aspect that should be considered probably even more than the roughness). Maybe the authors could add a discussion in relation to other literature findings for similar materials.

2) From Fig. S5 one can see that after 30 cycles of abrasion tests the coating is partially removed. So, I maintain the opinion expressed in my first report that the contact angles measured after several cycles can be assigned to glass support (contact angle not provided, although suggested). I strongly recommend rephrasing this part and the conclusions of these tests.

Reviewer 3 Report

The authors addressed most of my questions. However, there is one question for clarification.

The wetting behavior of PAA, being soluble in water, may seem peculiar to study since prolonged exposure to water could dissolve the coating. However, even though the quantity of PAA is minimal, it still provides coherence to the silica layer. The low power law coefficient is likely attributable to the hydrophilic nature of silica, as well as increasing amounts of PAA. It may be worth considering the hydrophilicity of silica, rather than just its roughness, when discussing these results. In light of this, the authors might want to discuss other literature findings on comparable materials.

Author Response

We thank the reviewer for the careful reading and thoughtful feedback.

We agree that PAA is water-soluble. However, dissolving process of PAA coating by water was expected to take much more than several seconds.[1] In this work, the entire experimental time was around 5 seconds.  Additionally, the molecular weight of the PAA was 450,000 and the solubility would be low in this time frame.  (This was one of the reasons for choosing this type of PAA) Given such a short time, as well as the high MW PAA and low PAA content in the final coating composition, it is reasonable to believe that the effect of the water-soluble property on the dynamic wetting behavior is negligible.

We agree with reviewer that the very low power law coefficient is likely related to water adsorption on PAA, rather than roughness. We made this statement in the revised manuscript which reads as:

“This volume decrease is attributed to the adsorption of water by PAA due chemical and thermodynamic interactions.”

 “Droplet volume decrease was considered to be the main reason for the low power law index values.”

[1] Han, T., Ma, Z., & Wang, D. (2021). Biofouling-inspired growth of superhydrophilic coating of polyacrylic acid on hydrophobic surfaces for excellent anti-fouling. ACS Macro Letters, 10(3), 354-358.